

# Continuous Néel-VBS quantum phase transition in non-local one-dimensional systems with SO(3) symmetry

Chao-Ming Jian[1], Yichen Xu[2], Xiao-Chuan Wu[2] and Cenke Xu[2]

**1** Station Q, Microsoft, Santa Barbara, California 93106-6105, USA
**2** Department of Physics, University of California, Santa Barbara, CA 93106, USA

## Abstract

One dimensional ($1d$) interacting systems with local Hamiltonians can be studied with various well-developed analytical methods. Recently novel $1d$ physics was found numerically in systems with either spatially nonlocal interactions, or at the $1d$ boundary of $2d$ quantum critical points, and the critical fluctuation in the bulk also yields effective nonlocal interactions at the boundary. This work studies the edge states at the $1d$ boundary of $2d$ strongly interacting symmetry protected topological (SPT) states, when the bulk is driven to a disorder-order phase transition. We will take the $2d$ Affleck-Kennedy-Lieb-Tasaki (AKLT) state as an example, which is a SPT state protected by the SO(3) spin symmetry and spatial translation. We found that the original $(1+1)d$ boundary conformal field theory of the AKLT state is unstable due to coupling to the boundary avatar of the bulk quantum critical fluctuations. When the bulk is fixed at the quantum critical point, within the accuracy of our expansion method, we find that by tuning one parameter at the boundary, there is a generic direct transition between the long range antiferromagnetic Néel order and the valence bond solid (VBS) order. This transition is very similar to the Néel-VBS transition recently found in numerical simulation of a spin-1/2 chain with nonlocal spatial interactions. Connections between our analytical studies and recent numerical results concerning the edge states of the $2d$ AKLT-like state at a bulk quantum phase transition will also be discussed.

Our understanding of one dimensional ($1d$) quantum many-body systems with local Hamiltonians is far more complete compared with higher dimensional systems, since many powerful analytical methods such as Bethe ansatz [1], Virasoro algebra [2], etc. are applicable only to $1d$ systems (or $(1+1)d$ space-time). We also understand that $1d$ systems have many unique features that are fundamentally different from higher dimensions. For example, with local Hamiltonians, generally there can not be spontaneous continuous symmetry breaking in $(1+1)d$ even at zero temperature (with exceptions of the scenarios when a fully polarized ferromagnet is the exact ground state), the closest one can possibly get is a quasi-long range power-law correlation of order parameters that transform nontrivially under a continuous symmetry. There is also no topological order in $1d$ systems analogous to fractional quantum Hall states which have a gap and simultaneously ground state topological degeneracy [3]. This means that many phenomena that are found in higher dimensions do not occur in $1d$ systems.

To seek for richer physics in one dimensional systems, we need to explore beyond the restriction of local Hamiltonians. One way to get around this restriction is to consider $1d$ systems at the boundary of a $2d$ systems, and drive the $2d$ bulk to a quantum phase transition. The physics becomes especially interesting when the disordered phase in the phase diagram of the $2d$ bulk is a symmetry protected topological (SPT) phase, which already has topologically protected $1d$ edge state. The interplay between the topological edge state and gapless quantum critical modes can lead to very nontrivial physics, which has been studied through numerical methods recently [4–7]. One can also directly turn on nonlocal spatial interaction in a $1d$ Hamiltonian. $1d$ quantum spin chains with nonlocal spatial interactions have also been studied recently, and very intriguing physics was found [8, 9]. We will discuss the results of these numerical works later in this paper.

In this work we investigate the $2d$ SPT state protected by symmetry $SO(3) \times G$, where $SO(3)$ is the ordinary spin symmetry, while $G$ is a discrete symmetry, which could be an onsite unitary $Z_2$ symmetry, or an anti-unitary time-reversal $Z_2^T$. $G$ can also be a lattice symmetry such as translation by one lattice constant. For example, when $G$ is the translation along the $\hat{x}$ axis ($T_x$), this state can be realized as the Affleck-Kennedy-Lieb-Tasaki (AKLT) state of the spin-2 system on a $2d$ square lattice [10]. In the example of spin-2 AKLT state, there is a chain of dangling spin-$1/2$ at the boundary of the system, as long as the boundary is along the $\hat{x}$ axis and preserves the translation symmetry $T_x$. The nature of the SPT states, and the Lieb-Shultz-Mattis (LSM) theorem [11–13] guarantee that this boundary system cannot be trivially gapped, *i.e.* it must be either gapless, or gapped but degenerate (For a closed $1d$ system without $0d$ boundaries, a generic ground state degeneracy can only originate from spontaneous discrete symmetry breaking [3]). In this work we will take the AKLT state as an example, but our results can be straightforwardly generalized to other discrete symmetries $G$.

Our study will mainly focus on the $1d$ boundary of strongly interacting $2d$ bosonic SPT phases, using a controlled renormalization group method. We would like to mention that previous literature has discussed the coupling between quantum criticality and topologically localized gapless states in various fermionic topological insulators [14]; other approaches such as constructing soluble models and various numerical methods have also been used to study edge states of interacting SPT states at a bulk quantum criticality [15–17]. Our main finding is that there is a generic continuous quantum phase transition between a long range antiferromagnetic Néel order which spontaneously breaks the $SO(3)$ spin symmetry, and a valence bond solid state, at the $1d$ boundary of an AKLT state that couples to the bulk quantum critical modes. The bulk quantum critical modes effectively yield nonlocal interactions at the $1d$ boundary, which makes the long range Néel order possible.

In principle the $1d$ boundary of this AKLT state should be effectively described by an extended Heisenberg model

$$H = \sum_j J\vec{S}_j \cdot \vec{S}_{j+1} + \cdots, \tag{1}$$

where $\vec{S}_j$ is the spin-1/2 operator, and the ellipsis includes other possible terms allowed by $SO(3) \times T_x$. The ground state of Eq. 1 depends on the entire lattice Hamiltonian. But a useful starting point of analyzing this boundary system is the $SU(2)_1$ conformal field theory (CFT) described by the following Hamiltonian in the infrared limit:

$$H_0 = \int dx \, \frac{1}{3 \cdot 2\pi} \left( \vec{J}_L \cdot \vec{J}_L + \vec{J}_R \cdot \vec{J}_R \right). \tag{2}$$

The $SU(2)_1$ CFT has a larger symmetry than the lattice Hamiltonian Eq. 2, since $\vec{J}_L$ and $\vec{J}_R$ generate the $SU(2)_{L,R}$ symmetries for the left and right chiral modes respectively. The relation

between the microscopic operator $\vec{S}$ and the low energy field is [18]

$$\vec{S}(x) \sim \frac{1}{2\pi} \left( \vec{J}_L(x) + \vec{J}_R(x) \right) + (-1)^x \vec{n}(x), \tag{3}$$

where $\vec{n}(x)$ is the Néel order parameter at the boundary. $\vec{J}_{L,R}$ both have scaling dimension $+1$ at the $SU(2)_1$ CFT fixed point, while $\vec{n}(x)$ has scaling dimension $1/2$ at the $SU(2)_1$ CFT.

The diagonal $SU(2)$ symmetry (simultaneous $SU(2)$ rotation between the left and right modes) corresponds to the original $SO(3)$ spin symmetry on the lattice scale. And because the lattice Hamiltonian has a lower symmetry than the infrared theory Eq. 2, another term is allowed in the low energy Hamiltonian:

$$H_1 = \int dx\, \lambda \vec{J}_L \cdot \vec{J}_R. \tag{4}$$

Since $\vec{J}_{L,R}$ have scaling dimension $+1$, power-counting indicates the coefficient $\lambda$ has scaling dimension $0$. Depending on the sign of $\lambda$, this term can be either marginally relevant or marginally irrelevant. When $\lambda$ is negative and marginally irrelevant the system flows back to the $SU(2)_1$ CFT with an enlarged $SU(2)_L \times SU(2)_R$ symmetry. When this term is positive and marginally relevant, it will flow to infinite (nonperturbative) and generate a mass gap, which based on the nature of the SPT phase would imply that the system spontaneously breaks the discrete symmetry $G$. For example, when this system is realized as the AKLT state, and $G$ is the translation $T_x$, the LSM theorem demands that when the boundary of the system generates a mass gap, it spontaneously breaks the translation symmetry and develops a nonzero expectation value of a dimerized valence bond solid (VBS) order: $\nu \sim (-1)^j \vec{S}_j \cdot \vec{S}_{j+1}$. As a side-note, we emphasize that the state we are studying here is different from the $SO(3)$ or $SU(2)$ SPT state defined through the group cohomology of $SO(3)$ or $SU(2)$ [19–21], since in those states the symmetry acts chirally, *i.e.* it only acts on either the left or right modes. While in our case the spin symmetry acts on both the left and right modes of the $1d$ boundary, and another discrete symmetry such as translation is demanded.

Our goal is to study the edge states when the bulk undergoes a disorder-order quantum phase transition, and the disordered phase of the bulk phase diagram is the AKLT state. The quantum critical fluctuation in the bulk may affect the edge of the AKLT state. To study the interplay between the topologically protected edge states, and the quantum critical modes, we adopt the "two layer" picture used in Ref. [22]: in layer-1, the system remains a gapped AKLT state in the bulk with solid edge states described by Eq. 1 and Eq. 2; in layer-2 the system undergoes a phase transition between an ordinary *trivial* disordered phase and an ordered phase. These two systems are glued together at the boundary. We have used the common wisdom that the transition between the SPT phase and the ordered phase is generically in the same universality class as the transition between an ordinary disordered phase and an ordered phase [1]. We will discuss two kinds of ordered phases: an $SO(3)$ antiferromagnetic order, and an Ising-like VBS order that spontaneously breaks $T_x$, assuming the boundary is at $y = 0$. In the bulk the two disorder-order transitions under discussion correspond to the three dimensional ($3D$) $SO(3)$ and Ising Wilson-Fisher transitions respectively, which can be studied through a standard $\epsilon = 4 - D$ expansion, where $D = 2 + 1$ is the space-time dimension in the bulk. We only extend the bulk dimensionality of layer-2 to $3 - \epsilon$ spatial dimensions, while the layer-1 still has a two-dimensional bulk and one-dimensional boundary.

---

[1]This statement can be inferred based on the observation that, the topological effects of many of the SPT states can be captured by a nonlinear Sigma model plus a topological $\Theta$−term at $\Theta = 2\pi$ [23, 24]. The $\Theta = 2\pi$ topological term reduces precisely to a boundary term, and we do not expect this topological term to change the bulk universality class.

We denote the bulk SO(3) antiferromagnetic order parameter, and the Ising-VBS order parameter in layer-2 as $\vec{\phi}$ and $\phi$ respectively, which should couple to the Néel order parameter $\vec{n}$ and the VBS order parameter $v$ at the boundary theory of layer-1, and this coupling could lead to new physics in the infrared. However, $\vec{\phi}$ and $\phi$ do not directly couple to $\vec{n}$ and $v$ due to the boundary condition of the Wilson-Fisher fixed point. Assuming the boundary of the 2$d$ system is at $y = 0$, the most natural boundary condition for fields $\vec{\phi}, \phi$ would be $\vec{\phi}(y = 0) = \phi(y = 0) = 0$ [2]. Then the leading nonvanishing boundary fields with the same quantum number as $\vec{\phi}$ and $\phi$ are $\vec{\Phi} \sim \partial_y \vec{\phi}$ and $\Phi \sim \partial_y \phi$ [25].

The SO(3) order parameter $\vec{\phi}$ and the Ising order parameter $\phi$ will not become critical simultaneously without fine-tuning, but they can be treated in the same framework. The boundary quantum critical modes $\vec{\Phi}$ and $\Phi$ couple to the fields at the boundary of layer-1 through the following terms in the action

$$
\begin{aligned}
\mathcal{S} \;=\; & \int d^2\mathbf{x}\, g_n \vec{\Phi}(\mathbf{x}) \cdot \vec{n}(\mathbf{x}) + g_v \Phi(\mathbf{x}) v(\mathbf{x}) \\
& + \int d^2\mathbf{x} d^2\mathbf{x}'\, \frac{1}{2} \Phi^a(\mathbf{x}) C_n^{-1}(\mathbf{x}, \mathbf{x}')_{ab} \Phi^b(\mathbf{x}') \\
& + \int d^2\mathbf{x} d^2\mathbf{x}'\, \frac{1}{2} \Phi(\mathbf{x}) C_v^{-1}(\mathbf{x}, \mathbf{x}') \Phi(\mathbf{x}'),
\end{aligned}
\tag{5}
$$

where $\mathbf{x} = (x, \tau)$ is the space-time coordinate. $C_n(\mathbf{x}, \mathbf{x}')_{ab}$ and $C_v(\mathbf{x}, \mathbf{x}')$ are the normalized correlation functions of $\Phi^a$ and $\Phi$ at the boundary:

$$
\begin{aligned}
C_n(\mathbf{x}, 0)_{ab} &= \langle \Phi^a(x, \tau) \Phi^b(0, 0) \rangle = \frac{\delta_{ab}}{(x^2 + \tau^2)^{3/2 - \epsilon_n}}, \\
C_v(\mathbf{x}, 0) &= \langle \Phi(x, \tau) \Phi(0, 0) \rangle = \frac{1}{(x^2 + \tau^2)^{3/2 - \epsilon_v}}.
\end{aligned}
\tag{6}
$$

The scaling dimension of $\vec{\Phi}$ and $\Phi$ is $\Delta_n = D/2 - \epsilon_n + O(\epsilon^2)$ and $\Delta_v = D/2 - \epsilon_v + O(\epsilon^2)$, where $D = 3$ is the bulk space-time dimension. $\epsilon_{n/v}$ can be computed again through the $\epsilon = (4 - D)$ expansion, following the calculation of boundary criticality of the Wilson-Fisher fixed points [25–29]: for an O($N$) Wilson-Fisher fixed point in the bulk, the scaling dimension of the boundary modes of the order parameter is

$$
\Delta_{O(N)} = \frac{D}{2} - \frac{N+2}{2(N+8)} \epsilon + O(\epsilon^2).
\tag{7}
$$

In our case $\epsilon_{n/v} = \epsilon(N + 2)/(2(N + 8))$ with $N = 3, 1$ respectively. We again stress that the $\epsilon$ dimensionality was introduced for layer-2 only. The effective action of $\vec{\Phi}$ and $\Phi$ in Eq. 5 already received leading order correction from the $\epsilon$−expansion due to the self-interaction of the bulk critical modes. These effective actions can in principle receive further corrections from the $g_v$ and $g_n$ couplings with the boundary fields $\vec{n}$ and $v$, but this correction should be at least at the order of $g_n^2, g_v^2$, which will be at higher order of $\epsilon$−expansion. As we can see later, the main physics we will discuss is at the vicinity of a fixed point where $g_n, g_v \sim \epsilon$.

Eq. 2, 4, 5 together can be viewed as an effective non-local 1$d$ theory, and this theory will be the starting point of our discussion hereafter. Considering the fact that the scaling dimension of both the Néel and VBS order parameter at the SU(2)$_1$ CFT is 1/2, to the leading order of $\epsilon$ expansion, the scaling dimensions of the coupling constants must be

$$
\Delta_{g_n} = \epsilon_n + O(\epsilon^2), \quad \Delta_{g_v} = \epsilon_v + O(\epsilon^2)
$$

---

[2] This boundary condition corresponds to the "ordinary transition" in the standard boundary criticality literatures; other possibilities can also occur such as special and extraordinary boundary transitions [25].

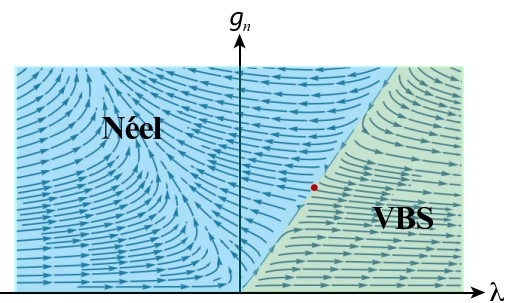

Figure 1: The coupled RG flow of $\lambda$ and $g_n$ based on Eq. 11. A new fixed point $(\lambda^*, g_n^*) = (\frac{2\epsilon_n}{\pi}, \frac{4\epsilon_n}{\pi})$ is found, which separates two phases: the phase where $\lambda \to +\infty$ is the VBS phase, and the phase with $(\lambda, g_n) \to (-\infty, +\infty)$ is the long range Néel order at the $1d$ boundary. But on the Néel order side of the phase diagram, the RG flow is complicated and nonmonotonic, hence it may take a long RG scale, or a large system size to finally reveal the true long range order.

$$\epsilon_n = \frac{5}{22}\epsilon, \quad \epsilon_v = \frac{1}{6}\epsilon. \tag{8}$$

$g_{n/v}$ are hence weakly relevant assuming a small parameter $\epsilon$. Hence the SU(2)$_1$ CFT at the boundary of the AKLT state will be unstable against coupling to the quantum critical modes, while fortunately due to the weak relevance of the coupling constants, this effect can be studied perturbatively.

To proceed we need to compute the coupled renormalization group (RG) flow of $\lambda$ and $g_{n/v}$ in Eq. 4 and Eq. 5. The RG equations can be derived based on the following operator product expansion (OPE):

$$J_L^a(z)n^b(w,\bar{w}) \sim \frac{1}{2}\frac{1}{z-w}\left(\mathrm{i}\delta_{ab}v(w,\bar{w}) + \mathrm{i}\epsilon_{abc}n^c(w,\bar{w})\right),$$

$$J_R^a(\bar{z})n^b(w,\bar{w}) \sim \frac{1}{2}\frac{1}{\bar{z}-\bar{w}}\left(-\mathrm{i}\delta_{ab}v(w,\bar{w}) + \mathrm{i}\epsilon_{abc}n^c(w,\bar{w})\right),$$

$$J_L^a(z)v(w,\bar{w}) \sim -\frac{1}{2}\frac{\mathrm{i}}{z-w}n^a(w,\bar{w}),$$

$$J_R^a(\bar{z})v(w,\bar{w}) \sim \frac{1}{2}\frac{\mathrm{i}}{\bar{z}-\bar{w}}n^a(w,\bar{w}).$$

$$\left(\sum_a n^a(z,\bar{z})\Phi^a(z,\bar{z})\right)\left(\sum_b n^b(w,\bar{w})\Phi^b(w,\bar{w})\right)$$

$$\sim \frac{3}{2}\frac{1}{|z-w|^4} + \frac{1}{2}\frac{1}{|z-w|^2}\sum_{a=1,2,3}J_L^a(w)J_R^a(\bar{w})$$

$$+ \frac{3}{4}\frac{1}{(\bar{z}-\bar{w})^2}T_L(w) + \frac{3}{4}\frac{1}{(z-w)^2}T_R(\bar{w}) + ...,$$

$$(v(z,\bar{z})\Phi(z,\bar{z}))(v(w,\bar{w})\Phi(w,\bar{w}))$$

$$\sim \frac{1}{2}\frac{1}{|z-w|^4} - \frac{1}{2}\frac{1}{|z-w|^2}\sum_{a=1,2,3}J_L^a(w)J_R^a(\bar{w})$$

$$+ \frac{1}{4}\frac{1}{(\bar{z}-\bar{w})^2}T_L(w) + \frac{1}{4}\frac{1}{(z-w)^2}T_R(\bar{w}) + ...,$$

$$\left(\sum_{a=1,2,3} J_L^a(z)J_R^a(\bar{z})\right)\left(\sum_{b=1,2,3} J_L^b(w)J_R^b(\bar{w})\right)$$

$$\sim \frac{3}{4}\frac{1}{|z-w|^4} - \frac{2}{|z-w|^2}\sum_{a=1,2,3} J_L^a(w)J_R^a(\bar{w})$$

$$+\frac{3}{2}\frac{1}{(\bar{z}-\bar{w})^2}T_L(w) + \frac{3}{2}\frac{1}{(z-w)^2}T_R(\bar{w}) + ... \tag{9}$$

In these equations, $z$ and $w$ are the chiral coordinates ($z = \tau + ix$); and the ellipsis contains less singular terms of the OPEs. The fields $T_{L/R}$ are the energy-momentum tensor of the left and right movers, which are given via the Suguwara construction by $T_L = \frac{1}{3}\sum_a : J_L^a J_L^a :$ and $T_R = \frac{1}{3}\sum_a : J_R^a J_R^a :$. Notice the form of energy-momentum tensors is similar to the Hamiltonian Eq. 2 but with an extra factor of $2\pi$. The OPEs above involving the fields $\Phi^a$ and $\Phi$ are derived to the leading order of $\epsilon_{n/v}$.

These OPEs are sufficient to derive the desired RG equations to the second order of the coupling constants. For example, using the first two lines of Eq. 9, we can derive another set of secondary OPEs:

$$\left(\sum_{a=1,2,3} J_L^a(z)J_R^a(\bar{z})\right)\left(\sum_b n^b(w,\bar{w})\Phi^b(w,\bar{w})\right)$$

$$\sim \frac{1}{4}\frac{1}{|z-w|^2}\left(\sum_b n^b(w,\bar{w})\Phi^b(w,\bar{w})\right),$$

$$\left(\sum_{a=1,2,3} J_L^a(z)J_R^a(\bar{z})\right)(v(w,\bar{w})\Phi(w,\bar{w}))$$

$$\sim -\frac{3}{4}\frac{1}{|z-w|^2}(v(w,\bar{w})\Phi(w,\bar{w})). \tag{10}$$

The coupled RG equations (beta functions) for $\lambda$ and $g_{n/v}$ then read

$$\beta(\lambda) = \frac{d\lambda}{d\ln l} = 2\pi\lambda^2 - \frac{\pi}{2}g_n^2 + \frac{\pi}{2}g_v^2,$$

$$\beta(g_n) = \frac{dg_n}{d\ln l} = \epsilon_n g_n - \frac{\pi}{2}\lambda g_n,$$

$$\beta(g_v) = \frac{dg_v}{d\ln l} = \epsilon_v g_v + \frac{3\pi}{2}\lambda g_v. \tag{11}$$

These RG equations are valid as long as we restrict our analysis to the parameter region with $\lambda, g_n, g_v \sim \epsilon$, since every term in the RG equations Eq. 11 would be at the same order of $\epsilon^2$.

As we explained before, there is no general reason for $\vec{\phi}, \phi$ to become critical simultaneously in the bulk. Hence let us ignore the $\Phi$ field first, and consider the coupled RG equation for $\lambda, g_n$ only. If there is no bulk quantum critical modes, an initial positive value $\lambda = \lambda_0$ will be marginally relevant, and open up an energy gap when it flows to positive infinite. According to the LSM theorem, and the nature of the SPT state, this $1d$ boundary cannot be trivially gapped, hence a nonperturbative positive $\lambda$ would drive the system into an SO(3) invariant VBS state with spontaneous symmetry breaking of translation symmetry $T_x$. But by coupling to the boundary modes $\vec{\Phi}$ of quantum critical fluctuation, the beta functions have an new unstable fixed point at

$$(\lambda^*, g_n^*) = \left(\frac{2\epsilon_n}{\pi}, \frac{4\epsilon_n}{\pi}\right). \tag{12}$$

The two eigenvectors of RG flow expanded at the new fixed point have scaling dimensions $(8.9\epsilon_n, -0.89\epsilon_n)$.

Of course the RG analysis above is only at the leading nontrivial order of $\epsilon$−expansion, and at this order of accuracy, no other fixed point is found in the phase diagram. The new fixed point found above separates two phases: phase I where $\lambda$ flows to positive infinity, and phase II where $\lambda$ and $g_n$ flow to negative and positive infinity respectively. Then both phases no longer have scaling invariance, so both phases should have certain long range order considering the fact that there is no topological order in one dimension [3]. Phase I with $\lambda \to +\infty$ is the dimerized VBS phase as we discussed before; phase II with $(\lambda, g_n) \to (-\infty, +\infty)$ should be a Néel ordered phase, *i.e.* the $1d$ boundary can develop the Néel order before the bulk, even though the bulk is still at a quantum critical point. A negative $\lambda$ would enhance the correlation of the Néel order parameter, and after integrating out $\vec{\Phi}$, a long range interaction proportional $g^2$ would be generated between the Néel order parameters. Hence the infrared limits $\lambda \to -\infty$ and $g \to +\infty$ of phase II both favor the long range Néel order.

The correlation length critical exponent $\nu$ of this Néel-VBS transition is $\nu \sim 1/(8.9\epsilon_n)$. At the transition point $(\lambda^*, g_n^*) = (2\epsilon_n/\pi, 4\epsilon_n/\pi)$, the scaling dimensions of the Néel and VBS order parameters can again be computed to the leading order of $\epsilon$−expansion:

$$
\begin{aligned}
\Delta_{\vec{n}} &= \frac{1}{2} + \frac{\pi\lambda^*}{2} = \frac{1}{2} + \epsilon_n, \\
\Delta_{\nu} &= \frac{1}{2} - \frac{3\pi\lambda^*}{2} = \frac{1}{2} - 3\epsilon_n.
\end{aligned}
\tag{13}
$$

One can see that compared with the $SU(2)_1$ CFT, the Néel order correlation is suppressed while the VBS order correlation is enhanced at the new transition fixed point, since $\lambda^* > 0$. This also implies that this Néel-VBS transition has no enlarged symmetry of $SU(2)_L \times SU(2)_R$. An enlarged $SU(2)_L \times SU(2)_R \sim SO(4)$ symmetry would guarantee that the Néel and VBS order parameters have the same scaling dimension, because $(\vec{n}, \nu)$ transform as a vector under $SO(4)$. Many previous studies suggest that at an unconventional quantum critical point between two phases with different spontaneous symmetry breaking, an enlarged emergent symmetry in the infrared is often expected due to a series of dualities [30–36]. But in our current case we expect the infrared symmetry at the Néel-VBS transition is still the microscopic symmetry $SO(3) \times G$.

As we mentioned before, suppose we integrate out the field $\vec{\Phi}$ in Eq. 5, a long range interaction in space-time will be generated between the Néel order parameter. The scenario is similar to the spin-1/2 chain with a long range spin-spin interaction, the only difference is that in the latter case the long range interaction is instantaneous and only nonlocal in space. Recently a direct transition between the Néel and VBS order was found in a spin-1/2 chain with nonlocal two-spin interaction and local four-spin interaction [8, 9]. It was found numerically that at the direct Néel-VBS transition the scaling dimension of the Néel order parameter is greater than the VBS order parameter, which is fundamentally different from the $SU(2)_1$ CFT, but consistent with our RG calculations Eq. 13. We also note that a previous RG analysis was performed for $1d$ spin-1/2 system with an instantaneous nonlocal spin interaction, but the Néel-VBS transition was not found therein. Instead the previous analysis identified a transition between the true long range Néel order and a quasi-long range order at the parameter region $\epsilon_n < 0$ and $\lambda < 0$ with our notation [37].

So far we have assumed that the fields $\vec{n}, \nu$ and $\vec{\Phi}, \Phi$ have the same velocity in our effective $1d$ theory Eq. 5, hence the theory we considered so far has a Lorentz invariance. We can also turn on a weak velocity difference between these two sets of fields, and analyze how it flows under RG. This velocity anisotropy corresponds to modifying the correlation function of $\vec{\Phi}$:

$$
C_n(\mathbf{x}, 0)_{ab} = \langle \Phi^a(x, \tau) \Phi^b(0, 0) \rangle
$$

$$= \frac{\delta_{ab}}{\left((1 - \frac{\delta v}{2})^2 x^2 + (1 + \frac{\delta v}{2})^2 \tau^2\right)^{3/2}}. \tag{14}$$

Here we have assumed that the velocity of $\vec{\Phi}$ exceeds the velocity of $\vec{n}$ by a factor of $(1 + \delta v)$ (to the first order of $\delta v$). We have taken $\epsilon_n = 0$ for the leading order calculation. $\delta v$ can flow under RG as it is the "seed" for velocity difference. Based on symmetry, the RG flow of $\delta v$ should look like

$$\frac{d\delta v}{d \ln l} = -\alpha g_n^2 \delta v. \tag{15}$$

And eventually we will plug in the fixed point value of $g_n = g_n^*$. Based on previous experience, at an interacting fixed point, a weak velocity anisotropy is often irrelevant [38, 40], since intuitively in the infrared all the interacting modes are expected to have the same velocity. Hence we expect $\alpha > 0$, *i.e.* a weak velocity difference between the boundary and bulk will be irrelevant at the Néel-VBS transition fixed point.

To evaluate $\alpha$, we expand the correlation function of $\vec{\Phi}$ to the leading order of $\delta v$:

$$C_n(\mathbf{x}, 0) = \frac{1}{|z|^3} - \frac{3}{2} \frac{\delta v}{|z|^5} \frac{z^2 + \bar{z}^2}{2} + O(\delta v^2). \tag{16}$$

Using the OPEs in Eq. 10, the second order perturabtion of $g_n$ would generate the following term:

$$-\frac{1}{2} g_n^2 \left( \sum_a n^a(z, \bar{z}) \Phi^a(z, \bar{z}) \right) \left( \sum_b n^b(w, \bar{w}) \Phi^b(w, \bar{w}) \right)$$
$$\sim -\frac{3g_n^2}{4|z - w|^4} - g_n^2 \frac{1}{4} \frac{1}{|z - w|^2} \sum_{a=1,2,3} J_L^a(w) J_R^a(\bar{w})$$
$$+ g_n^2 \delta v \frac{9}{32} \frac{1}{|z - w|^2} (T_L(w) + T_R(\bar{w})) + \cdots \tag{17}$$

Here we only kept the terms that will lead to nonzero effect under real space RG. The last term in Eq. 17 would contribute a renormalization (or acceleration) for the velocity of $\vec{n}$. Under rescaling, the ratio between the two velocities reduces by a factor:

$$1 + \delta v \to \frac{1 + \delta v}{1 + g_n^2 \delta v \frac{9\pi^2}{8} \ln l}, \tag{18}$$

which leads to the RG equation for $\delta v$:

$$\frac{d\delta v}{d \ln l} = -\frac{9\pi^2}{8} (g_n^*)^2 \delta v, \tag{19}$$

which confirms our expectation that $\delta v$ is an irrelevant perturbation at the Néel-VBS transition fixed point.

Suppose we start with $\delta v > 0$, namely the velocity of $\vec{n}$ is smaller than $\vec{\Phi}$, the velocity of $\vec{n}$ will increase under RG. This means that in this case the system will qualitatively behave like $z < 1$, where $z$ is the dynamic critical exponent (not to confuse with the chiral coordinate). On the contrary, if we start with $\delta v < 0$, the velocity of $\vec{n}$ would decrease under RG, which means that effectively $z > 1$. The former scenario is analogous to a spin chain with instantaneous spatial nonlocal interaction [9], which is equivalent to taking the velocity of the effective action of $\vec{\Phi}$ and $\Phi$ to infinity in our effective $1d$ theory Eq. 5. Although our calculation is for $\delta v > 0$, rather than taking the velocity in the $\vec{\Phi}$ action to be infinity, the "acceleration" of the modes

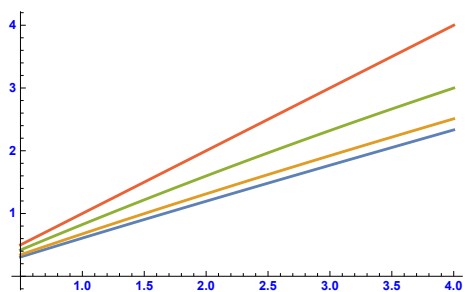

Figure 2: The plot of $\ln[3\pi G_n(\mathbf{k})(1+A(g_n^{*\prime})^2)]$ against $\ln[1/|\mathbf{k}|]$, where $G_n(\mathbf{k})$ is given by Eq. 20. From top to bottom, $A(g_n^{*\prime})^2 = 0, 1/2, 2$, and $5$.

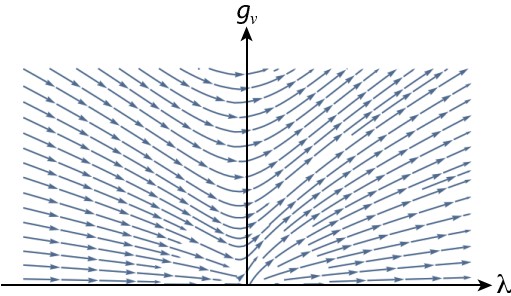

Figure 3: The RG flow of $(\lambda, g_v)$. As long as the initial value $g_v$ is nonzero, both parameters will flow to positive infinity, which implies that the boundary will likely develop the Ising-VBS order before the bulk.

derived here (including $z < 1$) is qualitatively consistent with what was observed in Ref. [9] at the Néel-VBS transition in a spin-1/2 chain with nonlocal spatial interactions.

In the phase diagram Fig. 1, on the side of the Néel order, the path of the RG flow towards the long range order can be complicated. It may take a long RG scale and hence large system size to reveal the true long range order. For example, on part of the phase diagram, $\lambda$ changes its sign and eventually flow away to the negative nonperturbative regime. While $\lambda$ changes sign, $g_n$ first decreases its magnitude from the initial value $g_0$, then after reaching its minimum $g_n^{*\prime}$ along the RG flow, $g_n$ keeps increasing and eventually become nonperturbative. Hence it is possible that for a relatively large intermediate scale, the system behaves like $g_n \sim g_n^{*\prime}$. The effect of this nonmonotonic RG flow can be illustrated by a simple perturbation theory to the correlation function of the Néel order parameter:

$$G_n(\mathbf{x}) = \langle \vec{n}(\mathbf{x}) \cdot \vec{n}(0) \rangle$$

$$\sim \frac{3}{2}\frac{1}{|\mathbf{x}|} + \frac{3}{4}\int d^2\mathbf{x}_1 d^2\mathbf{x}_2 \frac{(g_n^{*\prime})^2}{|\mathbf{x}-\mathbf{x}_1||\mathbf{x}_1-\mathbf{x}_2|^{3-2\epsilon_n}|\mathbf{x}_2|}$$

$$+ \ O(g_n^{*\prime})^4 + \cdots. \tag{20}$$

Hence $G_n(\mathbf{k})$ in the momentum-frequency space $\mathbf{k} = (k, \omega)$ reads

$$G_n(\mathbf{k}) \sim \frac{1}{G^{(0)}(\mathbf{k})^{-1} - \Sigma(\mathbf{k})}, \tag{21}$$

where $G^{(0)}(\mathbf{k}) = 3\pi/|\mathbf{k}|$, $\Sigma(\mathbf{k}) = -A(g_n^{*\prime})^2|\mathbf{k}|^{1-2\epsilon_n}/(3\pi)$, and $A > 0$ for $0 < \epsilon_n < 1/2$. The system will have enhanced spin-spin correlation function compared with the $SU(2)_1$ CFT of the spin-1/2 chain, as was observed in numerical simulations [4,6,7]. The mixture of the two terms in $G^{-1}(\mathbf{k})$ may yield results that appear to be power-law correlation with different scaling

dimensions, which is illustrated in Fig. 2, where we have fixed $\epsilon_n = 5/22\epsilon$ but chosen different $g_n^{*\prime}$. This nonuniversal power-law like scaling of spin correlation was also observed in recent numerics concerning the edge states of the AKLT state during a bulk phase transition [6,7].

Now we briefly consider the situation when the bulk undergoes a disorder-order quantum phase transition between the AKLT state and the Ising like VBS order, which is described by order parameter $\phi$. The boundary mode of $\phi$ is $\Phi \sim \partial_y \phi$, and it couples to the VBS order parameter $v$ at the boundary CFT. In this case, the coupled RG flow of $\lambda$ and $g_v$ in Eq. 5 is relatively simple: as long as we start with nonzero $(\lambda_0, g_{v0})$, both $g_v$ and $\lambda$ quite generally flow to positive infinity, which corresponds to a nonzero long range order of $v$. Hence the $1d$ boundary of the system should develop the Ising-VBS order before the bulk. when the bulk is tuned closer and closer to a VBS (Ising) transition, the boundary will go through a transition between the gapless $SU(2)_1$ CFT state to a VBS phase, before the bulk actually hits criticality. This boundary transition should be in the same universality class as the transition from an $SU(2)_1$ CFT to a VBS phase in a purely one-dimensional spin-1/2 chain with both nearest and next nearest neighbor Heisenberg interactions (see, for example, Ref. [39] for the one-dimensional transition). We note that this transition is not an ordinary $1+1d$ Ising transition and, hence, is different from the "extraordinary transition" studied in the standard boundary criticality literature. But if we start with a negative initial value $\lambda_0$, it may take a long RG time before the coupling constants become positive and nonperturbative. Hence the VBS order parameter may still appear to have quasi long range correlation for a finite system.

In conclusion, we have found that there can be a direct continuous quantum phase transition between the long range antiferromagnetic Néel order, and the VBS order, in an effective $1d$ spin-1/2 system with nonlocal interactions (Eq. 5). Due to the nonlocality of the model, even in a $1d$ system with a continuous SO(3) spin symmetry there can be a long range Néel order. Within the accuracy of our method, the effective spin-1/2 system Eq. 5 arises from coupling the $1d$ boundary of a $2d$ SPT phase to bulk quantum critical modes. Our results were drawn from a controlled renormalization group study, and the critical exponents extracted (including the anomalous dimensions of order parameters and the dynamical exponent) are qualitatively consistent with the Néel-VBS transition found numerically in recent simulation of a spin-1/2 chain with spatially instantaneous nonlocal interactions [8,9]. If a $1d$ system has local interactions only, there can only be spontaneous discrete symmetry breaking. Previous numerical and analytical works [41–43] have studied the analogue of deconfined quantum critical point between two phases that spontaneously break different discrete symmetries.

**Funding information** This work is supported by NSF Grant No. DMR-1920434, the David and Lucile Packard Foundation, and the Simons Foundation. The authors thank Anders Sandvik and Leon Balents for helpful discussions.

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
