# Peer review of "Continuous N\'{e}el-VBS Quantum Phase Transition in Non-Local one-dimensional systems with SO(3) Symmetry"

_SciPost Physics, doi:SciPost Phys. 10, 033 (2021)_

## Round 1 · Referee Report · Anonymous (Referee 1) · 2020-7-24

Strengths

Interesting question, solid analysis.

Weaknesses

The discussion in the introduction lumps together non-locality in space with non-locality in spacetime:
In the case where we integrate out gapless bulk modes, the effective action is nonlocal in both time and space.
This is distinct from the case of a hamiltonian with long-range interactions, which is the subject of the simulations
quoted in the introduction. At this early stage in the study of such systems, it is not clear
how important this distinction is and I think it should not be glossed over.
Certainly this makes it difficult to take too seriously the (qualitative) comparisons with numerical results.

(I see that the authors study the effects of a perturbation in the velocity,
and that in the extreme limit, this removes the non-locality in time.
However, this analysis is only to leading order in small deviations from $v=c$.)

Please see "Requested Changes" for more comments.

Report

The study of gapless states with topological properties has been flourishing recently. This paper is a solid
addition to this literature and should be published.

Actually, although it is not written this way at all,
it might be more accurate to say that the paper is an addition to the literature on surface critical phenomena.
Certainly it is easier to understand in that context.

The authors are doing something very similar to what is done in their reference 14
(which for some reason has so far not been followed up as vigorously as it should be):
they are studying a 1+1d CFT (which happens to be protected by some symmetries) that sits at the boundary of a 2+1d critical point.
For most of the paper, they are studying the specific case of the $SU(2)_1$ CFT coupled to the SO(3) Wilson-Fisher theory in bulk.

The epsilon expansion is used to get perturbative control over the interactions.
Here $\epsilon = 3-d$ where $d$ is the bulk space dimension.
Is the (codimension one) boundary of a $(d-\epsilon)$-dimensional system still one dimensional?
In the usual RG treatment of surface critical phenomena by epsilon expansion,
I believe one varies the dimension of the boundary as well,
whereas here the authors are simultaneously using techniques of 1+1d CFT.
I guess it seems to work out.

Requested changes

-- I found this sentence confusing:
"As a side-note, we emphasize that the state we are studying here is different from the SO(3) or SU(2) SPT state in the standard notation [19– 21], since in those states the symmetry acts chirally, i.e. it only acts on either the left or right modes."

This is not just a question of notation, since the action of the symmetry is physically meaningful.
If I understand correctly, the authors are distinguishing between the 2d SU(2) AKLT state, which is a weak SPT (i.e. relies on translation symmetry for its stability) and an SU(2) SPT built using (some continuous-group version of) group cohomology,
where the symmetry acts chirally. Perhaps the authors can rephrase this sentence to make it clearer.
I think the word "standard" is not appropriate because of the fact that SU(2) is a continuous group.

-- I also found this description of the strategy confusing:
"in layer-1, the sys- tem remains a gapped AKLT state in the bulk with solid edge states described by Eq. 1 and Eq. 2; in layer-2 the system undergoes a phase transition between an ordinary trivial disordered phase and an ordered phase. These two systems are glued together at the boundary. "
This makes it sound like the two layers are only identified at the boundary.
If I understand correctly, this strategy relies on the claim that the order/disorder phase transitions in the AKLT system
is in the same universality class as the ordinary Wilson-Fisher fixed points. I don't dispute the claim, but if this understanding is correct, it should be made explicit.

-- Related to this, the discussion of the choice of boundary conditions on the "layer 2" critical fields is too terse.
This choice of $\phi_\mu|_{bdy} = 0$ has crucial consequences for the leading scaling dimensions of all the operators,
by sticking in extra $y$ derivatives.
This is not the only choice that can be realized physically,
and I think the authors should explain a bit more (for example very briefly reviewing the usual story of surface critical phenomena) and at least comment on the consequences of other choices, even if they are multicritical.

-- In the brief discussion of coupling to the bulk Ising fixed point, the authors conclude that
the boundary VBS order will form first. This phenomenology is the same as that of the "extraordinary" transition (as described e.g. in Cardy's book). Is it actually different?
(i.e. could they actually be the same, and if not, could you tell the difference from observables?)
I suppose the difference is that the authors have extra surface degrees of freedom not required by the $Z_2$ extraordinary transition.
I think some comment about this would be helpful.

-- before equation (1) "be effectively describe by"

-- nonperturbative is misspelled in "it will flow to infinite (nonpertubrative)"

-- after equation (21) there is a broken reference to numerical simulations.

  • validity: -
  • significance: -
  • originality: -
  • clarity: -
  • formatting: -
  • grammar: -

Author:  Cenke Xu  on 2020-12-13  [id 1076]

(in reply to Report 1 on 2020-07-24)

Response to report 1:

We want to thank referee 1 for very careful reading of our manuscript, and also very helpful comments.

Response to general comments:

Yes we are only generalizing the bulk dimensionality of layer-2 to 3-epsilon, the boundary dimension of layer-1 (AKLT layer) is always kept at 1d.

Although the flow of velocity in our work was done perturbatively, the “acceleration” of the CFT modes, and the fact that z < 1, is indeed (and should be) qualitatively consistent with the 1d spin-chain with nonlocal interaction observed numerically. Also, the fact that we found the Neel order parameter has a larger scaling dimension than the VBS order parameter at the deconfined critical point is also consistent with what was observed numerically. Hence we believe at least at the qualitative level the comparison/connection between our theory and numerics is worth making.

Response to other points:

1, We have added more clarification about the difference between the SU(2) SPT state and AKLT state.

2, Yes we believe that the SPT-to-order state transition should be generally the same as the Wilson-Fisher transition. The SPT states can in many cases be described by a nonlinear-sigma model with a topological Theta term at Theta = 2Pi, and at Theta = 2Pi, the Theta term is reduced to boundary terms. We do not expect the Theta = 2Pi topological term to modify the universality class in the bulk.

3, We have added brief discussions about other choices of boundary conditions.

4, This is a very good question. Yes the situation we are discussing here is different from the extraordinary transition in the standard boundary criticality literature. The boundary of layer-1 cannot be a trivial disordered phase in principle, protected by topology; while the ordinary boundary system can enter a trivial disordered phase. Hence in our system, when the bulk is tuned closer and closer to a VBS (Ising) transition, the boundary will go through a transition between the “gapless spin chain phase” with SU(2)_1 CFT, to a VBS phase, before the bulk hits criticality. This boundary transition should be in the same universality class as the transition from an SU(2)_1 CFT to a VBS phase in a purely one-dimensional spin-1/2 chain with both nearest and next nearest neighbor Heisenberg interactions. This transition is not an ordinary 1+1d Ising transition and, hence, is different from the ``extraordinary transition" studied in the standard boundary criticality. We have explained this in the revised manuscript.

Other typos and broken references have been fixed.

---

## Round 1 · Referee Report · Anonymous (Referee 2) · 2020-10-7

Report

Recent numerical work (Refs 4-7) explored boundary criticality in 2D quantum magnets with effective spin-1/2 chains at the boundary. The present paper makes a renormalization group analysis of this problem, at the bulk critical point, combining known properties of an isolated 1D chain with an epsilon expansion for the bulk.

This is an interesting paper and provide a useful framework for discussing these models, even if the RG treatment is not completely controlled. I recommend publication in Scipost, after the following few issues have been addressed.

1) The authors should clarify the status of the basic approximations. The RG approach is referred to as “controlled”. What does this mean? An epsilon expansion is used for the bulk theory. So the approximation would be controlled at small epsilon. However, small epsilon does not make sense here because the boundary dimensionality is fixed. Is it possible to access a controlled limit?

2) Boundary condition. The authors say “the most natural boundary condition for fields φ⃗, φ would be φ⃗(y = 0) = φ(y = 0) = 0.” This is clear when the boundary is disordered. How to see it is appropriate everywhere in the phase diagram?

3) Equation (5). Are the lines 2 and 3 a gaussian approximation to the bulk theory? Is such an approximation justified? If so this should be explained.

4) Below fig 2 there is a comparison between the RG results and numerical results in Refs 4,5. This should be more clearly explained. Are the authors suggesting that simulations of Refs 4,5 are close to the boundary critical point but slightly inside boundary Neel phase?

5) It is stated “a spin-1/2 chain with spatially nonlocal interactions, which should be equivalent to our example by taking the velocity of the bulk quantum critical modes to infinity.” I do not think this is correct. In the fixed point studied, the boundary velocity renormalizes to become equal to the finite bulk velocity. A fixed point in which the bulk velocity is infinite seems fundamentally different.

  • validity: -
  • significance: -
  • originality: -
  • clarity: -
  • formatting: -
  • grammar: -

Author:  Cenke Xu  on 2020-12-13  [id 1077]

(in reply to Report 2 on 2020-10-07)

Response to report 2:

We want to thank referee 2 for very helpful comments.

1, We view our system as a two-layer system: layer-2 is an ordinary Wilson-Fisher critical point in the 2+1d bulk, and layer-1 is a SPT state in the 2+1d bulk with a nontrivial 1+1d boundary state. Then we couple the 1+1d boundary of layer-1 to layer-2. After introducing fractional dimensional epsilon, our system becomes 3-epsilon dimensional critical point (4-epsilon space-time dimension) from the bulk of layer-2, but still coupled to the 1+1d boundary of layer-1. We do not virtually generalize the spatial dimensionality of the layer-1 boundary, and we do not view the boundary of layer-1 as a 2-epsilon dimensional system. Hence the epsilon expansion is only for the layer-2, which is a standard Wilson-Fisher fixed point. And indeed for the Wilson-Fisher fixed point (and its boundary) the epsilon expansion is controlled. We have clarified this in our revised manuscript.

2, This boundary condition is for layer-2, which is always at the critical point in the bulk, hence there is no long range order in layer-2 at least in the bulk. Hence we believe it is fine to take the standard boundary condition of layer-2. The main physics we discuss is the novel CFT-like fixed point at the coupled 1+1d boundary, which also has no long range order.

Then our discussion is based on the vicinity of the novel fixed point at the coupled 1+1d boundary. If the layer-1 forms a long range order at the 1+1d boundary due to the runaway flow from the fixed point, this long range order can in principle induce an order at the boundary of layer-2 through a “back-reaction”. But this back-reaction will involve higher order effects of the coupling between the two layers, which should correspond to higher-order in the epsilon expansion, because the coupling between the two layers is at the order of epsilon, if we focus on the vicinity of the fixed point at the coupled 1+1d boundary.

3, Lines 2 and 3 in Eq.5 are not Gaussian approximation of the bulk theory. The anomalous dimension from the Wilson-Fisher fixed point is already accounted for in Eq.5 (to the order of epsilon), which means that Feynman diagrams of self-interaction of the order parameter from layer-2 are already taken into account in Eq.5. So line-2 and 3 should be valid to the leading order of epsilon. Then we take Eq.5 as the starting point of studying the interaction between the layer-2 and layer-1. There could be further corrections to the “effective actions” of the $\Phi$ fields in line-2 and 3 in Eq.5, but these corrections should correspond to higher order expansion of epsilon, either from higher order epsilon-expansion of the standard boundary theory of the Wilson-Fisher fixed point, or from correction from the coupling between the two layers at the boundary. Let us again keep in mind that the couplings between the two layers ($g_n$ and $g_v$ in Eq.5) are also at order of epsilon in the vicinity of the main fixed point we are dicussing, so if we include the corrections from $g_n$ and $g_v$ to the effective action of $\Phi$ fields in Eq.5, these corrections will be at least at the $g_n^2$ or $g_v^2$ order, which means $epsilon^2$ order.

4, Yes we want to explain the observation from numerics that the scaling at the AKLT boundary is not universal when the bulk is tuned to a critical point, i.e. the scaling at the boundary depends on the coupling constants at the boundary. Our interpretation is that the models used in the numerical references are in principle in the AFM ordered side of our RG flow diagram, but this long range order may emerge only at very large RG scale. So at different scale away from the fixed point (tunable by the boundary coupling constant) the system may appear to have different scaling behavior.

5, Indeed, more precisely we should have said that an instantaneous long range interaction between spins in the 1d spin-chain considered in Ref.8,9, are analogous to the effective 1d theory Eq.5 after taking the velocity of the effective action of \Phi to infinity. By the way here we view Eq.5 as an independent 1+1d theory, rather than a theory inferred from a bulk theory. We have clarified this in our revised manuscript. In our calculation we took the velocity of the effective action of $\Phi$ to be larger than the velocity of the original SU(2)_1 CFT, and indeed the velocity of the CFT increases under RG, and flow to the velocity of the effective action of $\Phi$ in Eq.5. This “acceleration” under RG will lead to dynamical exponent z < 1, which is qualitatively consistent with what was observed numerically.

---

## Round 2 · Referee Report · Anonymous (Referee 1) · 2021-1-7

Report

The authors have responded to all the comments and I think the paper should be published now.

---

## Round 2 · Referee Report · Anonymous (Referee 2) · 2021-2-7

Report

The paper is now ready for publication.

---

## Round 2 · Author Response

Dear editor,

We sincerely apologize for the delayed response. This year has been very stressful for many of us.

We want to thank both referees for their very careful reading of our paper, and their helpful questions/comments. Following is our response to the reports, and summary of changes according to each comment from the report.

Response to report 2:

1, We view our system as a two-layer system: layer-2 is an ordinary Wilson-Fisher critical point in the 2+1d bulk, and layer-1 is a SPT state in the 2+1d bulk with a nontrivial 1+1d boundary state. Then we couple the 1+1d boundary of layer-1 to layer-2. After introducing fractional dimensional epsilon, our system becomes 3-epsilon dimensional critical point (4-epsilon space-time dimension) from the bulk of layer-2, but still coupled to the 1+1d boundary of layer-1. We do not virtually generalize the spatial dimensionality of the layer-1 boundary, and we do not view the boundary of layer-1 as a 2-epsilon dimensional system. Hence the epsilon expansion is only for the layer-2, which is a standard Wilson-Fisher fixed point. And indeed for the Wilson-Fisher fixed point (and its boundary) the epsilon expansion is controlled. We have clarified this in our revised manuscript.

2, This boundary condition is for layer-2, which is always at the critical point in the bulk, hence there is no long range order in layer-2 at least in the bulk. Hence we believe it is fine to take the standard boundary condition of layer-2. The main physics we discuss is the novel CFT-like fixed point at the coupled 1+1d boundary, which also has no long range order. Then our discussion is based on the vicinity of the novel fixed point at the coupled 1+1d boundary. If the layer-1 forms a long range order at the 1+1d boundary due to the runaway flow from the fixed point, this long range order can in principle induce an order at the boundary of layer-2 through a “back-reaction”. But this back-reaction will involve higher order effects of the coupling between the two layers, which should correspond to higher-order in the epsilon expansion, because the coupling between the two layers is at the order of epsilon, if we focus on the vicinity of the fixed point at the coupled 1+1d boundary.

3, Lines 2 and 3 in Eq.5 are not Gaussian approximation of the bulk theory. The anomalous dimension from the Wilson-Fisher fixed point is already accounted for in Eq.5 (to the order of epsilon), which means that Feynman diagrams of self-interaction of the order parameter from layer-2 are already taken into account in Eq.5. So line-2 and 3 should be valid to the leading order of epsilon. Then we take Eq.5 as the starting point of studying the interaction between the layer-2 and layer-1. There could be further corrections to the “effective actions” of the Phi fields in line-2 and 3 in Eq.5, but these corrections should correspond to higher order expansion of epsilon, either from higher order epsilon-expansion of the standard boundary theory of the Wilson-Fisher fixed point, or from correction from the coupling between the two layers at the boundary. Let us again keep in mind that the couplings between the two layers (g_n and g_v in Eq.5) are also at order of epsilon in the vicinity of the main fixed point we are dicussing, so if we include the corrections from g_n and g_v to the effective action of \Phi fields in Eq.5, these corrections will be at least at the g_n^2 or g_v^2 order, which means epsilon^2 order.

4, Yes we want to explain the observation from numerics that the scaling at the AKLT boundary is not universal when the bulk is tuned to a critical point, i.e. the scaling at the boundary depends on the coupling constants at the boundary. Our interpretation is that the models used in the numerical references are in principle in the AFM ordered side of our RG flow diagram, but this long range order may emerge only at very large RG scale. So at different scale away from the fixed point (tunable by the boundary coupling constant) the system may appear to have different scaling behavior.

5, Indeed, more precisely we should have said that an instantaneous long range interaction between spins in the 1d spin-chain considered in Ref.8,9, are analogous to the effective 1d theory Eq.5 after taking the velocity of the effective action of \Phi to infinity. By the way here we view Eq.5 as an independent 1+1d theory, rather than a theory inferred from a bulk theory. We have clarified this in our revised manuscript. In our calculation we took the velocity of the effective action of \Phi to be larger than the velocity of the original SU(2)_1 CFT, and indeed the velocity of the CFT increases under RG, and flow to the velocity of the effective action of \Phi in Eq.5. This “acceleration” under RG will lead to dynamical exponent z < 1, which is qualitatively consistent with what was observed numerically.

Response to report 1:

Response to general comments:

Yes we are only generalizing the bulk dimensionality of layer-2 to 3-epsilon, the boundary dimension of layer-1 (AKLT layer) is always kept at 1d.

Although the flow of velocity in our work was done perturbatively, the “acceleration” of the CFT modes, and the fact that z < 1, is indeed (and should be) qualitatively consistent with the 1d spin-chain with nonlocal interaction observed numerically. Also, the fact that we found the Neel order parameter has a larger scaling dimension than the VBS order parameter at the deconfined critical point is also consistent with what was observed numerically. Hence we believe at least at the qualitative level the comparison/connection between our theory and numerics is worth making.

Response to other points:

1, We have added more clarification about the difference between the SU(2) SPT state and AKLT state.

2, Yes we believe that the SPT-to-order state transition should be generally the same as the Wilson-Fisher transition. The SPT states can in many cases be described by a nonlinear-sigma model with a topological Theta term at Theta = 2Pi, and at Theta = 2Pi, the Theta term is reduced to boundary terms. We do not expect the Theta = 2Pi topological term to modify the universality class in the bulk.

3, We have added brief discussions about other choices of boundary conditions.

4, This is a very good question. Yes the situation we are discussing here is different from the extraordinary transition in the standard boundary criticality literature. The boundary of layer-1 cannot be a trivial disordered phase in principle, protected by topology; while the ordinary boundary system can enter a trivial disordered phase. Hence in our system, when the bulk is tuned closer and closer to a VBS (Ising) transition, the boundary will go through a transition between the “gapless spin chain phase” with SU(2)_1 CFT, to a VBS phase, before the bulk hits criticality. This boundary transition should be in the same universality class as the transition from an SU(2)_1 CFT to a VBS phase in a purely one-dimensional spin-1/2 chain with both nearest and next nearest neighbor Heisenberg interactions. This transition is not an ordinary 1+1d Ising transition and, hence, is different from the ``extraordinary transition" studied in the standard boundary criticality. We have explained this in the revised manuscript.

---

## Round 2 · List of Changes

1, We further clarified our epsilon expansion. As was noted by referee-1, we are only generalizing the dimensionality of "layer-1", i.e. the layer with Wilson-Fisher criticality in the bulk. The boundary of the SPT phase is fixed at one dimension.

2, We added explanation about choosing the ordinary boundary condition of "layer-2", and briefly discussed other possible boundary conditions in the standard boundary criticality literature (in the footnote).

The reason for choosing this boundary condition, and the nature of the effective action of \Phi and \vec{\Phi} in Eq.5 were also further explained in the revised manuscript. The effective action is not a Gaussian approximation, it already contained self-interaction between the critical modes from the bulk, at least to the leading order of epsilon expansion. In principle it will receive further corrections, for example from the coupling to \vec{n}, but that will be higher order effect in the epsilon expansion, because this correction involves higher order effect of the coupling constant g_n. In our calculation, g_n, and g_v will be at order of epsilon, if we focus on the physics around the fixed point we found.

3, We have clarified our understanding of the velocity flow. We view Eq.2,4, and 5 together as an effective 1d theory, and what we meant was that, the instantaneous spatial interaction introduced in long range spin-chain, is equivalent to taking the velocity of the action of \vec{\Phi} to infinity in Eq.5. Although the flow of velocity in our work was done perturbatively, the “acceleration” of the CFT modes, and the fact that z < 1, is indeed (and should be) qualitatively consistent with the 1d spin-chain with nonlocal interaction observed numerically.

Also, the fact that we found the Neel order parameter has a larger scaling dimension than the VBS order parameter at the deconfined critical point is also consistent with what was observed numerically. Hence we believe at least at the qualitative level the comparison/connection between our theory and numerics is worth making.

4, We have added more clarification about the difference between the SU(2) SPT state and AKLT state.

5, As was suggested by referee-1, we added text and a footnote to explain that we believe the SPT-to-ordered state phase transition, is in the same universality class as the ordinary disorder-to-order phase transition.

6, We have explained that, indeed, when the boundary forms VBS order before the bulk in our case, the transition at the boundary is different from the standard "extra-ordinary transition" in boundary criticality literature.

Other typos and broken references have been fixed.

---

## Editorial Decision

published